# In Vitro and In Vivo Evaluation of 6-*O*-α-Maltosyl-β-Cyclodextrin as a Potential Therapeutic Agent Against Niemann-Pick Disease Type C

**DOI:** 10.3390/ijms20051152

**Published:** 2019-03-06

**Authors:** Nushrat Yasmin, Yoichi Ishitsuka, Madoka Fukaura, Yusei Yamada, Shuichi Nakahara, Akira Ishii, Yuki Kondo, Toru Takeo, Naomi Nakagata, Keiichi Motoyama, Taishi Higashi, Yasuyo Okada, Junichi Nishikawa, Atsushi Ichikawa, Daisuke Iohara, Fumitoshi Hirayama, Katsumi Higaki, Kousaku Ohno, Muneaki Matsuo, Tetsumi Irie

**Affiliations:** 1Department of Clinical Chemistry and Informatics, Graduate School of Pharmaceutical Sciences, Kumamoto University, 5-1 Oe-honmachi, Chuo-ku, Kumamoto 862-0973, Japan; nushratyasmin@gmail.com (N.Y.); 171y2002@st.kumamoto-u.ac.jp (M.F.); 162y3101@st.kumamoto-u.ac.jp (Y.Y.); 126p1031@st.kumamoto-u.ac.jp (S.N.); 161p2001@st.kumamoto-u.ac.jp (A.I.); ykondo@kumamoto-u.ac.jp (Y.K.); 2Program for Leading Graduate Schools “HIGO (Health life science: Interdisciplinary and Glocal Oriented) Program”, Kumamoto University, 5-1 Oe-honmachi, Chuo-ku, Kumamoto 862-0973, Japan; 3Division of Reproductive Engineering, Center for Animal Resources and Development (CARD), Kumamoto University, 2-2-1 Honjo, Kumamoto 860-0811, Japan; takeo@kumamoto-u.ac.jp (T.T.); nakagata@gpo.kumamoto-u.ac.jp (N.N.); 4Department of Physical Pharmaceutics, Graduate School of Pharmaceutical Sciences, Kumamoto University, 5-1 Oe-honmachi, Chuo-ku, Kumamoto 862-0973, Japan; motoyama@gpo.kumamoto-u.ac.jp (K.M.); higashit@kumamoto-u.ac.jp (T.H.); 5Institute Biosciences, School of Pharmacy and Pharmaceutical Sciences, Mukogawa Women’s University, 11-68 Koshien Kyuban-cho, Nishinomiya 663-8179, Japan; okada@mukogawa-u.ac.jp (Y.O.); nisikawa@mukogawa-u.ac.jp (J.N.); aichikaw@mukogawa-u.ac.jp (A.I.); 6Laboratory of Physical Pharmaceutics, Faculty of Pharmaceutical Sciences, Sojo University, 4-22-1 Ikeda, Nishi-ku, Kumamoto 860-0082, Japan; dio@ph.sojo-u.ac.jp (D.I.); fhira@ph.sojo-u.ac.jp (F.H.); 7Division of Functional Genomics, Research Center for Bioscience and Technology, Faculty of Medicine, Tottori University, 86 Nishi-cho, Yonago 683-8503, Japan; kh4060@med.tottori-u.ac.jp; 8Sanin Rosai Hospital, 1-8-1, Kaikeshinden, Yonago 683-8605, Japan; ohno@sanmedia.or.jp; 9Department of Pediatrics, Faculty of Medicine, Saga University, 5-1-1, Nabeshima, Saga 849-8501, Japan; matsuo@cc.saga-u.ac.jp

**Keywords:** Niemann-Pick disease type C, cyclodextrin, 6-*O*-α-maltosyl-β-cyclodextrin, 2-hydroxypropyl-β-cyclodextrin

## Abstract

Niemann-Pick disease Type C (NPC) is a rare lysosomal storage disease characterized by the dysfunction of intracellular cholesterol trafficking with progressive neurodegeneration and hepatomegaly. We evaluated the potential of 6-*O*-α-maltosyl-β-cyclodextrin (G2-β-CD) as a drug candidate against NPC. The physicochemical properties of G2-β-CD as an injectable agent were assessed, and molecular interactions between G2-β-CD and free cholesterol were studied by solubility analysis and two-dimensional proton nuclear magnetic resonance spectroscopy. The efficacy of G2-β-CD against NPC was evaluated using *Npc1* deficient Chinese hamster ovary (CHO) cells and *Npc1* deficient mice. G2-β-CD in aqueous solution showed relatively low viscosity and surface activity; characteristics suitable for developing injectable formulations. G2-β-CD formed higher-order inclusion complexes with free cholesterol. G2-β-CD attenuated dysfunction of intercellular cholesterol trafficking and lysosome volume in *Npc1* deficient CHO cells in a concentration dependent manner. Weekly subcutaneous injections of G2-β-CD (2.9 mmol/kg) ameliorated abnormal cholesterol metabolism, hepatocytomegaly, and elevated serum transaminases in *Npc1* deficient mice. In addition, a single cerebroventricular injection of G2-β-CD (21.4 μmol/kg) prevented Purkinje cell loss in the cerebellum, body weight loss, and motor dysfunction in *Npc1* deficient mice. In summary, G2-β-CD possesses characteristics favorable for injectable formulations and has therapeutic potential against in vitro and in vivo NPC models.

## 1. Introduction

Niemann-Pick disease type C (NPC) is a recessive disorder caused by mutations of the *NPC1* (~95% of patients) or *NPC2* gene [1]. Transmembrane NPC1 and soluble luminal NPC2 proteins interact with unesterified free cholesterol (FC) and have an important role in intracellular cholesterol trafficking from the late endosome/lysosome to other organelles including the endoplasmic reticulum [2,3]. Cellular insufficiency such as cholesterol trafficking dysfunction (FC accumulation and shortage of esterified cholesterol (EC)) increases lysosomal volume, and autophagy malfunction occurs in patients with NPC, causing NPC-related manifestations including hepatosplenomegaly and central nerve dysfunction. NPC is a fatal hereditary intractable disease and the development of an effective cure is urgently required.

Recent studies have shown that 2-hydroxypropyl-β-cyclodextrin (HP-β-CD), a cyclic oligosaccharide, solubilized exogenous and endogenous lipophilic molecules, including cholesterol, to normalize intracellular cholesterol trafficking as an artificial cholesterol shuttle and prolong the lifespan of *Npc1* deficient (*Npc1*^−/−^) mice [4,5,6,7]. Although HP-β-CD is an effective drug in NPC patients, recent studies reported issues related to its physicochemical properties [8]. HP-β-CD contains a partially substituted poly (2-hydroxpropyl) ether of β-cyclodextrin and the 2-hydroxypropyl groups are randomly substituted onto the hydroxyl groups of the glucopyranose units. Therefore, an exact molecular weight cannot be determined because of the different degrees of substitution (D.S.) within a mixture. Indeed, the formulations of HP-β-CD (VTS-270 (Kleptose HPB) and Trappsol^®^ Cyclo™) used in clinical trials have different D.S. and molecular weights [8,9]. These properties of HP-β-CD are problematic during the development of a clinical formulation. Therefore, there is still a need to generate superior cyclodextrin derivatives other than HP-β-CD for the treatment of NPC.

6-*O*-α-maltosyl-β-cyclodextrin (G2-β-CD) is a reaction product of maltose and β-CD produced by *Pseudomonas* isoamylase (EC 3.2.1.68) (Figure 1). The primary hydroxyl group of β-cyclodextrin in G2-β-CD is substituted by maltose through an α-1,6 glycosidic linkage [10]. G2-β-CD is a mono-substituted derivative and has an exact molecular weight of 1459.27 [11]. The chemical purity of an active ingredient is an important element during the research and development of injectable formulations for clinical use. G2-β-CD can be obtained in a high state of purity [12], which is a great advantage over HP-β-CD, as the latter is an amorphous mixture of chemically-related components with different degrees of substitution. Previous studies suggested that G2-β-CD strongly reduced cellular cholesterol levels by forming inclusion complexes, with low cytotoxicity, in various cultured cells [13,14,15]. We previously reported that G2-β-CD was internalized in *Npc1* deficient Chinese hamster ovarian (CHO) cells and that it reached the lysosome [16]. Although these findings suggest that G2-β-CD might have therapeutic potential for NPC, few studies have reported the effect of G2-β-CD on NPC manifestations observed in cellular and animal models.

Based on these facts, we evaluated the physicochemical properties (viscosity and surface activity in aqueous solution) of G2-β-CD, intended for use as an injectable ingredient against NPC, and the molecular interactions between G2-β-CD and FC by solubility analysis and 2-dimensional proton nuclear magnetic resonance (2D-^1^H-NMR) spectroscopy. In addition, we examined the effect of G2-β-CD on the dysfunction of intracellular cholesterol trafficking and lysosome volume expansion in *Npc1* deficient CHO cells. We also evaluated the effect of G2-β-CD on hepatic and neuronal disorders in *Npc1* deficient mice when administered subcutaneously or intracerebroventricularly.

## 2. Results

### 2.1. Physicochemical Characterization of G2-β-CD Intended for Use as an Injectable Ingredient

When cyclodextrin derivatives are intended for clinical use as an injectable ingredient against NPC, especially administered by intracerebroventricular or intrathecal injection, a high viscosity and surface activity of the aqueous solution should be considered potential hazardous factors. Therefore, we measured the viscosity and surface tension of G2-β-CD solution (300 mM) in ultra-pure water and compared it with the HP-β-CD solution, the concentration of which was chosen for intracerebroventricular injection in this study. As shown in Figure 2, the viscosity of both β-CDs decreased gradually with an increase in rate of shear, showing a feature of quasi-viscous flow. The viscosity of G2-β-CD solution was significantly lower than that for the HP-β-CD solution.

The surface tensions of both β-CD solutions are shown in Table 1. Although a significant change was observed in the surface tension of the HP-β-CD solution compared with ultra-pure water, there was no significant difference between the G2-β-CD solution and ultra-pure water.

### 2.2. Evaluation of the Molecular Interactions of G2-β-CD with FC by Solubility Analysis and 2D-^1^H-NMR Spectroscopy

To elucidate the molecular interaction of G2-β-CD with FC, we measured the FC solubilizing ability of G2-β-CD in aqueous medium [17] and compared it with the HP-β-CD solution. The phase solubility diagram for FC with G2-β-CD and HP-β-CD in cell culture medium (pH 7.4, 37 °C) is shown in Figure 3. The solubility of FC increased exponentially with an increase in G2-β-CD concentration. The diagram is of the Ap type, which indicates the formation of higher-order soluble complexes [18]. The FC solubilizing potential of G2-β-CD was comparable to that in a previous study (Okada et al.) and was equal to or greater than with HP-β-CD. We also showed the phase solubility was 2–10 mM for G2-β-CD in Appendix A.

We used 2D-^1^H-NMR spectroscopy to determine the mode of interaction between G2-β-CD and FC in aqueous solution. The partial contour plots of the Rotating frame Overhauser effect spectroscopy (ROESY) spectrum of the FC:G2-β-CD system in D_2_O are shown in Figure 4. Cross peaks connecting the intermolecular protons were observed between the protons numbered 4, 7, 12, 18,19, 21, 26, and 27 in FC and the proton numbered 3 in the β-CD structure in G2-β-CD. In addition, weak cross peaks were observed between the protons numbered 18, 19, 21, 26, and 27 in FC and the protons numbered 5 or 6 in G2-β-CD. However, there was no significant cross peak between the protons in FC and the protons numbered 3′ or 5′ and 4′ in the branched maltose moiety of G2-β-CD. The full scale 2D-^1^H-NMR ROESY spectrum of G2-β-CD and G2-β-CD + FC solution is shown in Appendix A.

### 2.3. Attenuating Effects of G2-β-CD on Intracellular Cholesterol Trafficking Dysfunction and Lysosomal Volume Expansion in Npc1 Deficient Cells

To examine whether G2-β-CD has therapeutic potential against NPC pathology, we evaluated the effects of G2-β-CD on intracellular cholesterol levels and lysosome volume in *Npc1* deficient CHO cells using a previously reported method [19]. To evaluate the effects on cholesterol trafficking, we also separately measured intracellular FC and EC levels in *Npc1* deficient cells, which showed higher FC and lower EC levels than in WT cells. As shown in Figure 5A, treatment with G2-β-CD concentration-dependently decreased FC levels. Although the EC reduction was also attenuated by 0.1–1 mM of G2-β-CD, 4–10 mM had no attenuating effect (Figure 5B). The intracellular total cholesterol (TC) level is shown in Appendix A. The lysosomal volume evaluated by LysoTracker^®^ fluorescence intensity was significantly increased in *Npc1* deficient cells compared with WT cells. G2-β-CD also reduced the fluorescence intensity. Similar to the intracellular EC, the reducing effect appeared to diminish with 8–10 mM G2-β-CD compared with 0.1–1 mM (Figure 5C). We confirmed no significant cytotoxicity of G2-β-CD up to 10 mM in WT and *Npc1* deficient cells (Appendix A).

### 2.4. Hepatoprotective Effects of Subcutaneous G2-β-CD Treatment in Npc1 Deficient Mice

We evaluated the effect of the systemic treatment of G2-β-CD on hepatic changes in male and female homozygous mutant (BALB/cNctr-*Npc1*^m1N^, *Npc1*^−/−^) mice [20], used as an NPC model. As shown in Figure 6A, using *Npc1*^−/−^ mice, the G2-β-CD-treatment group had significantly reduced liver/body weight ratio compared with the saline-treated group. Representative histological pictures (H&E stain) of *Npc1*^−/−^ mouse livers are shown in Figure 6B. Vacuolated hepatocytes and Kupffer cells were observed in *Npc1*^−/−^ mice treated with saline. Of note, G2-β-CD treatment (2.9 mmol/kg, subcutaneous, once a week from 6 to 8 weeks of age) markedly inhibited these changes. G2-β-CD treatment also reduced FC content and tended to increase the EC/FC ratio in *Npc1*^−/−^ mouse livers (Figure 6C). Saline-treated *Npc1*^−/−^ mice showed abnormal serum transaminase levels (approximately 600 and 700 IU/L for alanine aminotransferase (ALT) and aspartate aminotransferase (AST), respectively), whereas these elevated transaminase levels were significantly attenuated by G2-β-CD treatment (Figure 6D). In addition, TUNEL analysis indicated higher numbers of hepatic parenchymal cells showed nuclear DNA fragmentation in saline treated *Npc1*^−/−^ mice compared with G2-β-CD treatment (Figure 6E).

We also measured the serum cholesterol level after subcutaneous treatment with G2-β-CD. A significant decrease in FC and a tendency for decreased TC levels was observed in the G2-β-CD-treated group compared with the saline-treated group (Appendix A). In addition, to evaluate the neuroprotective effect of subcutaneous G2-β-CD treatment, we examined changes in the immunohistostaining of calbindin, a marker of Purkinje cells, in the cerebellum after the subcutaneous treatment of G2-β-CD in *Npc1*^−/−^ mice. Although subcutaneous G2-β-CD significantly decreased the serum ALT and AST levels, no significant change was observed in the number of cerebellar calbindin-positive cells in these mice (Appendix A).

### 2.5. Neuroprotective Effects of Intracerebroventricular G2-β-CD Treatment in Npc1 Deficient Mice

Next, we examined the effect of an intracerebroventricular injection of G2-β-CD on the neuronal disorders observed in *Npc1*^−/−^ mice. As shown in Figure 7A, significant body weight loss was observed in saline-treated *Npc1*^−/−^ mice at 7–8 weeks of age. An intracerebroventricular injection of G2-β-CD (21.4 µmol/kg) at 4 weeks of age significantly prevented these changes. We also evaluated histological changes in the cerebellum of *Npc1*^−/−^ mice by calbindin-immunostaining (Figure 7B). Although only a few calbindin-positive cells were visible in saline treated *Npc1*^−/−^ mice at 8.3 weeks, many calbindin-positive cells were observed in the G2-β-CD treatment group (Figure 7C).

To further evaluate the neuroprotective effects in *Npc1*^−/−^ mice, we performed coat hanger and cage lid tests, used for the representative evaluation of motor function [21,22,23]. As shown in Figure 8A, dropping from the coat hanger was observed from 8 weeks of age in saline-treated *Npc1*^−/−^ mice. Statistically significant differences were observed for the hanging time in the coat hanger test between saline- and G2-β-CD-treated groups of *Npc1*^−/−^ mice at 8–10 weeks of age. The same tendency was also observed in the cage lid test, and the hanging time of the G2-β-CD-treated group was significantly prolonged compared with the saline-treated group in *Npc1*^−/−^ mice aged 10 weeks (Figure 8B).

To further examine the preventive effect of an intracerebroventricular injection of G2-β-CD, we evaluated the histological changes in whole *Npc1*^−/−^ mouse brain by osteoactivin/glycoprotein non-metastatic protein B (GPNMB)-immunostaining, a potential marker of NPC manifestation and lysosomal dysfunction [24,25]. GPNMB-positive cells were observed in the cerebellum and thalamus of saline-treated *Npc1*^−/−^ mice. However, little staining was observed in G2-β-CD treated mice (Appendix A).

## 3. Discussion

In this study, we demonstrated that G2-β-CD attenuated the dysfunction of intracellular cholesterol trafficking and increase in lysosomal volume in NPC model cells. The results of the FC solubility curve and 2D-^1^H-NMR analysis indicated that G2-β-CD interacts with FC at the concentration range that has attenuating effects in NPC model cells. In addition, we demonstrated that subcutaneous and intracerebroventricular treatment of G2-β-CD attenuated hepatic disorder and central nerve impairment in NPC model mice. These results suggest that G2-β-CD has therapeutic potential against NPC manifestations. Given the physicochemical properties of G2-β-CD including low viscosity and low surface activity of its solution, chemically pure G2-β-CD might be an attractive drug candidate for the future treatment of NPC.

The dysfunction of intracellular cholesterol trafficking in NPC cells occurs because of abnormal NPC1 and/or NPC2 proteins, and β-CD derivatives that potentially interact with cholesterol, such as the drug candidate HP-β-CD, might show attenuating effects against NPC [4,5,19]. The result of phase solubility curve analysis indicated that G2-β-CD interacted with FC in an aqueous solution within the effective concentration range (approximately 1-2 mM). This result is consistent with the result of experiments performed in serum-free Fisher’s medium as reported by Okada et al. [26]. The result of 2D-^1^H-NMR indicated that protons located around the interior of the cyclodextrin cavity that cross peaks are caused by the nuclear Overhauser effect, because they arise between two different molecules, suggesting close contact between the interacting protons. In addition, this result suggests that the secondary hydroxyl side of G2-β-CD is primarily directed to the dimethyl terminus of FC and includes the steroid skeleton in the case of the 1:1 molar complex. In the case of the 1:2 molar complex, the secondary hydroxyl side of the second G2-β-CD is expected to additionally direct to the hydroxyl terminal of FC. However, we cannot deny the possibility that a higher concentration of G2-β-CD/ FC may form an aggregation complex [27].

Regarding the in vitro efficacy evaluation, around 0.1–1.0 mM of G2-β-CD normalized the increase in FC levels and reduction in EC levels as well as the lysosomal volume in *Npc1* deficient cells. These results indicate the attenuation of the dysfunction of cholesterol and lysosomal trafficking. The attenuating effects of G2-β-CD tended to be reduced as the G2-β-CD concentration increased and 8–10 mM had little effect on abnormal cholesterol trafficking. This suggests that around 0.1–1.0 mM is the effective concentration range of G2-β-CD. In our previous study of HP-β-CD, we observed same phenomenon [19]. Therefore, when considering the administration plan of G2-β-CD or HP-β-CD in a clinical setting, dose- and concentration adjustments are important.

The subcutaneous treatment of *Npc1*^−/−^ mouse with G2-β-CD improved NPC-related liver dysfunctions, such as hepatocytomegaly, FC accumulation, and serum transaminase abnormalities. Hepatic evaluation demonstrated a large number of TUNEL stained cells, a marker for apoptotic cell death, in the NPC mouse liver, which were reduced by subcutaneous G2-β-CD administration. Although the exact mechanisms are still unclear, the results suggest that apoptotic hepatocellular injury occurred in the NPC liver and was reduced by subcutaneous G2-β-CD administration similar to that in FC abnormalities. The results of serum cholesterol measurement suggest that subcutaneous G2-β-CD administration also reduced serum cholesterol levels in *Npc1*^−/−^ mice. These results are consistent with the results of HP-β-CD reported by Ramirez et al. [6]. In addition, intracerebroventricular treatment with G2-β-CD prevented the reduction of calbindin-positive cells in the cerebellum, a measure of Purkinje cells, body weight and motor dysfunction indicating that intracerebroventricular G2-β-CD treatment protects against central nervous system (CNS) damage in the NPC mouse model. In our preliminary experiments, subcutaneous G2-β-CD treatment did not have any effect on the number of calbindin-positive cells in the cerebellum in contrast to its significant effects on serum transaminase levels in *Npc1*^−/−^ mice (Appendix A). Although further evaluation to confirm these findings are required, these results suggest that systemic treatment with G2-β-CD is less neuroprotective, similar to HP-β-CD, [28], and indicates the importance of intracerebroventricular treatment for the prevention of neuronal dysfunction.

When considering the development of a new drug formulation, particularly a parenteral formulation, the physicochemical property of the active ingredient and additive components, is important. In this study, we demonstrated that an aqueous solution of G2-β-CD had lower viscosity and surface activity compared with HP-β-CD solution. The concentration of the solutions was adjusted to be the same as solutions used for intracerebroventricular treatment. A high viscosity and surface activity causes problems for patients, including the force and needle bore size needed to administer a viscous injection, which may result in increased discomfort, user anxiety and irritation at the site of administration [29,30]. Because G2-β-CD is a mono-substituted β-CD derivative, its exact molecular weight can be determined. In contrast, it would be difficult to develop the HP-β-CD mono-substituted form because it would exist as a poly-substituted form with various D.S. values. Mono-substituted derivatives have the advantage that their exact concentration can be detected in serum or cerebrospinal fluid for pharmacokinetic analysis. Therefore, these physicochemical properties of G2-β-CD indicate it is an attractive drug candidate for this rare disease.

This study demonstrated that G2-β-CD had attenuating effects on NPC manifestations in model cells and mice and had favorable chemical purity and physicochemical properties. However, the study had some limitations. First, comparison of the attenuating effects of G2-β-CD with other CDs, including HP-β-CD, was not performed. Davidson et al. [31] suggested that sulfobutylether-β-CD and -γ-CD showed efficacy against NPC liver and brain manifestations with reduced ototoxicity in a comparative mouse study. Therefore, the comparison of G2-β-CD with other cyclodextrin (CD) derivatives regarding effectiveness and safety in model cells or animals is critical to confirm the positive effects of G2-β-CD. Second, the pharmacokinetic characterization of G2-β-CD, including its distribution characteristics, blood-brain barrier penetration, plasma protein binding, metabolism, elimination mode, therapeutic dosage, and blood concentration is still unclear. In this study, subcutaneous G2-β-CD did not reduce Purkinje cell loss in contrast to the intracerebroventricular treatment. The indirect preliminary results suggest systemic treatment with G2-β-CD has a little effect on CNS injury and that the penetration of G2-β-CD across the brood brain barrier of mice is low. Regarding HP-β-CD, Camargo et al. [32] suggested that it does not cross the blood-brain barrier in NPC mice. In addition, the dose of G2-β-CD used (2.9 mmol/kg for subcutaneous injection) was similar to that used for HP-β-CD [7,33] based on in vitro results. Although G2-β-CD attenuated NPC manifestations at this dose, its optimal dose is still unclear. Third, we only performed a short-term evaluation of hepatic and CNS injury in this animal study; the long-term efficacy in terms of life-prolonging effects is still unclear. In addition, only limited motor function tests, such as the coat hanger and cage lid tests, were performed in this study. To confirm motor function improvement by G2-β-CD, other performance tests related to NPC manifestations, such as the rotarod and footprint tests and quantitative magnetic resonance imaging of brain atrophy [34] should be conducted. Therefore, further studies to identify these factors are needed, although the in vivo results of the current study strongly suggest the attenuating potential of G2-β-CD against NPC. Fourth, the specific attenuating mechanisms of G2-β-CD were not fully demonstrated. Furthermore, the target molecule of other previously identified CD derivatives that had an effect on NPC, such as HP-β-CD, was considered to be FC. Indeed, Coisne et al. [35] demonstrated that β-CD and its derivatives, such as methylated-β-CD, trapped cellular cholesterol and altered the intracellular cholesterol pool/trafficking in smooth muscle and aortic endothelial cells. In this study, we demonstrated that G2-β-CD interacted with FC, and that G2-β-CD had the potential to interact with FC to an equal or greater degree than HP-β-CD; therefore G2-β-CD may attenuate the intracellular cholesterol metabolism and trafficking in NPC cells. To fully demonstrate the exact mechanisms, further study is warranted.

We report that G2-β-CD has therapeutic potential against in vitro and in vivo NPC models and surpasses HP-β-CD from the viewpoint of chemical purity and physicochemical properties favorable for injectable formulations, especially as an intrathecal or intracerebroventricular injection.

## 4. Materials and Methods

### 4.1. Reagents

G2-β-CD (MW: 1459.27, D.S.: 1) and HP-β-CD (average MW: 1402.38, D.S.: 4.61) were kindly donated by Ensuiko Sugar Refining Co., Ltd. (Tokyo, Japan) and Nihon Shokuhin Kako Co., Ltd. (Tokyo, Japan), respectively. Deuterium oxide (D_2_O) was purchased from Kanto Chemical Co., Inc. (Tokyo, Japan). Determiner L FC was purchased from Kyowa Medex Co., Ltd. (Tokyo, Japan). LysoTracker^®^ Green DND-26 and HyClone™ fetal bovine serum (FBS) were purchased from Thermo Fisher Scientific Inc. (Waltham, MA, USA). Dulbecco’s modified Eagle’s medium (DMEM) and F-12 medium were obtained from Gibco-Life Technologies (Life Technologies Japan, Tokyo, Japan). Four percent buffered paraformaldehyde (PFA) was purchased from Wako Pure Chemical Industries, Ltd. (Osaka, Japan). All other reagents and solvents were of reagent grade. Deionized and distilled biopure grade water was used throughout the study.

### 4.2. Measurement of Surface Tension

The surface tensions of G2-β-CD and HP-β-CD solutions (300 mM in ultra-pure water) were measured using a du Nouy tensiometer (Ito Seisakusho Co., Ltd., Tokyo, Japan) according to the du Nouy method [36,37]. Briefly, a platinum ring was lowered into the solutions being analyzed until completely submerged. Upon pulling the ring up and out of the solution, the force needed to break the contact of the ring with the solution is measured. We also measured the surface tension in HP-β-CD solution (300 mM) under the same conditions and compared this with G2-β-CD.

### 4.3. Measurement of Viscosity

The viscosity of G2-β-CD in solution was measured according to the method reported by Iohara et al. [38]. Briefly, we measured the rheological parameters of G2-β-CD solution (300 mM in ultra-pure water) using a MCR-101 rheometer (Anton Paar Japan K.K., Tokyo, Japan). A cone and plate geometry with a diameter of 25 mm and a 0.998 rad cone angle was used. Steady shear rheology measurements were monitored at 25 °C by increasing the shear rate from 1 to 100 s^−1^. We also measured the rheology in HP-β-CD solution (300 mM) under the same conditions and compared this with G2-β-CD.

### 4.4. Solubility Analysis

Solubility measurements were carried out according to our previous study with minor modifications [19]. An excess amount of FC (10 mg) was added to the culture media (DMEM:F12 = 1:1) containing various concentrations of G2-β-CD under aseptic conditions and shaken using the following parameters. 1: Shaking at room temperature for 60 min at 180 rounds/min; 2: shaking at 37 °C for 60 min at 120 rounds/min; 3: shaking at room temperature for 30 min at 180 rounds/min; and 4: shaking at 37 °C for 30 min at 120 rounds/min. After reaching equilibrium, the culture media were filtered through a Millex^®^-HP PES 0.45 μm (Merck Millipore Ltd., Cork, Ireland). The filtrates were mixed and shaken for 10 min with 2 mL of chloroform/methanol (2:1, *v/v*). After centrifugation, the chloroform phases were recovered and evaporated. The residues were dissolved in the solvent and the concentrations of FC were determined by Determiner L FC according to the manufacturer’s protocol.

### 4.5. Two-Dimensional Proton Nuclear Magnetic Resonance Spectroscopy

To examine the molecular interactions of G2-β-CD and FC, 2D-^1^H-NMR analysis was performed. An excess amount of cholesterol (30 mg) was added to D_2_O (2 mL) containing G2-β-CD (100 mM, 291.854 mg) under aseptic conditions and shaken using the following parameters. 1: Shaking at room temperature for 30 min at 180 rounds/min; and 2: shaking at 37 °C for 60 min (in a water bath) at 120 rounds/min. After reaching equilibrium, the suspension was filtered through a Millex^®^-HP PES 0.45 μm. Using the filtrate, the 2D-^1^H-NMR spectrum was recorded at 25 °C with BRUKER AVANCE-600 NMR (Bruker Co., Billerica, MA, USA). The spectrum in the rotating frame nuclear Overhauser effect spectroscopy (ROESY) mode was measured at 600 MHz and chemical shifts were given in parts per million (ppm) relative to that of tetramethylsilane (TMS).

### 4.6. Cellular Experiments

#### 4.6.1. Cell Culture

Cellular experiments were performed as we previously reported [19]. In brief, we used *Npc1* deficient CHO cells developed by Higaki et al. [39] as in vitro NPC model cells. Wild-type (WT) and *Npc1* deficient CHO cells were cultured in a 1:1 mixture of DMEM/F12 medium supplemented with 10% FBS, 100 IU/mL penicillin, 100 mg/mL streptomycin at 37 °C and 5% CO_2_. The cells were seeded on 96-, 12-, or 6-well plates, and 10 cm culture dishes and cultured for 24 h before use in various assays. We prepared an aqueous solution of G2-β-CD, a highly-water soluble β-CD mono substituent [13], without any organic solvents, such as dimethyl sulfoxide. The aqueous solution for the cellular experiment was prepared as followed. G2-β-CD powder was weighted and dissolved in DMEM/F12 medium using a 15 mL centrifuge tube. To completely solubilize it in the medium, it was vortexed and a 10 mL solution was prepared using a volumetric cylinder. The solution was sterilized by filtration and further diluted with DMEM/F12 medium prior to preparing different concentrations.

#### 4.6.2. Measurement of Intracellular Cholesterol Levels

WT or *Npc1* deficient cells were seeded in 6-well plates. The cells were treated with or without G2-β-CD for 24 h and then the cells were lysed. The protein concentration in the lysate was measured by a Pierce BCA Protein Assay Kit (Thermo Fisher Scientific Inc.) and then cholesterol was extracted from the lysate using chloroform, 2-propanol and an NP-40 mixture (7:11:0.1). The samples were centrifuged (15,000× *g* at 4 °C for 10 min) and collected the chloroform layers. The chloroform layers were evaporated and the residues were dissolved in 2-propanol, polyoxylene alkyl ether, and polyoxylene lauryl ether mixture (87:10:3). A portion of the solution was incubated with esterase to measure TC, and the other portion was incubated without esterase to measure FC. The cholesterol content in these samples was measured by Determiner L FC (Kyowa Hakko Kirin Co., Ltd., Tokyo, Japan) using a microplate reader (Tecan Co., Ltd., Männedorf, Switzerland). The EC level was calculated by deducting the FC level from the TC level. The intracellular cholesterol content was expressed as nmol/mg protein.

#### 4.6.3. Measurement of Lysosome Volume in Cells

Cells were exposed to medium containing G2-β-CD or medium without G2-β-CD for 24 h. Then the cells were stained with 50 nM LysoTracker^®^ Green DND-26 for 15 min at 37 °C and analyzed in a Flow Cytometer (BD Biosciences Accuri™ C6, Becton Dickinson Biosciences, Franklin Lakes, NJ, USA) using a 488-nm laser. The resulting fluorescence was detected in the FL1 channel using a 530 ± 30 nm filter. Data from 10,000 cells were collected and analyzed using accompanying software.

### 4.7. Animal Experiments

Male and female *Npc1*^−/−^ mice were bred and kept in specific pathogen-free conditions in the Center for Animal Resources and Development, Kumamoto University. Mice were housed in cages in a room under controlled conditions at 24 °C with a 12-h light cycle, and provided free access to food and water. All animal experiments were carried out at the Department of Clinical Chemistry and Informatics, Graduate School of Pharmaceutical Sciences, Kumamoto University. All experimental procedures conformed to the animal use guidelines of the Committee for Ethics on Animal Experiments of Kumamoto University (approval numbers A27-133 and A29-134, approval date: 1 April 2015 and 1 April 2017, respectively).

#### 4.7.1. Subcutaneous Administration of G2-β-CD into Npc1 Deficient Mice

Drug administration was performed according to our previous report of HP-β-CD [7]. In brief, 16 age-matched (6-weeks old) *Npc1*^−/−^ mice were divided into saline- and G2-β-CD-treated groups. The mice assigned to the saline group were treated with saline (19.18 µL/g) (*n* = 8; 4 males and 4 females), and mice assigned to the G2-β-CD group were treated with G2-β-CD (2.9 mmol/kg) (*n* = 8; 4 males and 4 females). We treated 6-week-old *Npc1*^−/−^ mice with filter-sterilized G2-β-CD solution or saline once a week until 8.3 weeks of age by subcutaneous injection (three injections in total). G2-β-CD solution for subcutaneous administration was prepared as followed. G2-β-CD powder was weighed and dissolved in half saline to maintain physiological osmolality (G2-β-CD concentration of the solution was 151.1 mM). We carefully vortexed it to completely solubilize it and the solution was prepared using a volumetric cylinder. After sterilization by filtration, we injected the solution into mice at a volume of 19.18 µL/g body weight.

#### 4.7.2. Evaluation of Hepatic Changes in Npc1 Deficient Mice

To examine the effect of subcutaneous G2-β-CD treatment on the hepatic manifestations observed in *Npc1*^−/−^ mice, we measured serum transaminase levels, body and liver weight changes, hepatic cholesterol content, and hepatic pathological changes. When *Npc1*^−/−^ mice in both groups reached 8.3 weeks of age, their body weight was measured and they were euthanized. Then blood and organ samples were immediately collected.

The blood samples were centrifuged at 3000× *g* at 4 °C for 10 min after coagulation and serum was collected for the measurement of aspartate aminotransferase (AST) and alanine aminotransferase (ALT) levels. Serum AST and ALT levels were determined using a bio-analyzer (SPOTCHEM EZ SP-4430; ARKRAY, Inc., Kyoto, Japan).

Liver samples were immediately weighed and then we estimated the ratio of liver weight to body weight as a measure of the enlargement of the liver. A portion of the hepatic lobes was immediately frozen in liquid nitrogen and stored at −80 °C for cholesterol measurements. Tissue homogenates were prepared in buffer (20 mM Tris, 2 mM ethylenediaminetetraacetic acid (EDTA), 150 mM NaCl, and 1% Triton X-100, pH 8) and smashed with Micro Smash™ MS-100R at 4 °C, 4000 rpm for 30 s three times. The solution was collected and aliquoted into two samples. One aliquot was incubated with cholesterol esterase at 37 °C for 30 min. The cholesterol content in these samples was measured by Determiner L FC (Kyowa Hakko Kirin Co., Ltd.). Other hepatic lobes were immediately fixed in 4% buffered paraformaldehyde (PFA) and then embedded in paraffin before being cut into 4-μm-thick sections for H&E staining and terminal deoxynucleotidyl transferase dUTP nick end labeling (TUNEL) assay. Staining for the TUNEL assay was performed using the ApopTag^®^ Peroxidase In Situ Apoptosis Detection Kit (Merck Millipore, Billerica, MA, USA) according to the manufacturer’s instructions. The liver histopathological changes were photographed and analyzed using a microscopic system (Biorevo BZ-9000; Keyence Co., Osaka, Japan). To quantify the TUNEL-stained cells, 18–20 random fields of each section were analyzed by microscopy in an open label manner, and quantification was performed by counting the number of positively-stained cells in each field.

#### 4.7.3. Intracerebroventricular Administration of G2-β-CD into Npc1 Deficient Mice

Eleven age-matched (4-weeks old) *Npc1^−/−^* mice were divided into saline- and G2-β-CD-treated groups. The mice assigned to the saline group were treated with saline (~1 µL/15 g body weight) (*n* = 6; 2 males and 4 females), and mice assigned to the G2-β-CD group were treated with G2-β-CD (21.4 µmol/kg) (*n* = 5; 3 males and 2 females). The preparation of the G2-β-CD solution for intracerebroventricular injection was as follows. G2-β-CD powder was weighted and dissolved in ultrapure water. We vortexed it to completely solubilize the powder and the volume of the solution was adjusted using a volumetric flask. The solution was prepared at a concentration of 320.8 mM and sterilized by filtration. A single cerebroventricular injection of G2-β-CD solution or saline into 4-week-old *Npc1^−/−^* mice was performed using a stereotaxic instrument for small animals (IMPACT-1000C and STEREOTAXIC INJECTOR KDS 310 plus; Muromachi Kikai Co., Ltd. Tokyo, Japan).

#### 4.7.4. Evaluation of Purkinje Cell Loss in the Cerebellum of Npc1 Deficient Mice

To evaluate the effect of intracerebroventricular G2-β-CD treatment on NPC neuronal dysfunction, we measured the amount of Purkinje cell loss in the cerebellum of mice. Mouse body weight was monitored until they were 8.3 weeks old and the mice were sacrificed after the final weight measurement. Then the mice were perfused with 4% buffered PFA through the aorta and the brain was isolated. The samples were fixed with 4% buffered PFA at 4 °C and embedded in paraffin. Microtome sections, 3-μm-thick, were incubated overnight at 4 °C with anti-calbindin D28K antibody (N-18) (Santa Cruz Biotechnology Inc., Santa Cruz, CA, USA; 1:100 dilution), a marker of Purkinje cells. The sections were also stained with Histofine^®^ SimpleStain MAX PO (Nichirei, Tokyo, Japan) and Mayer’s hematoxylin. The histological images were taken using a Biorevo BZ-9000 microscopic system (Keyence Co.) and calbindin-positive cells were counted in the whole cerebellum.

#### 4.7.5. Motor Function Tests in Npc1 Deficient Mice

To evaluate neuronal dysfunction in terms of behavior analysis, we performed motor function tests in *Npc1*^−/−^ mice. Twelve age-matched (4-weeks old) *Npc1*^−/−^ mice were divided into saline- and G2-β-CD-treated groups. The mice assigned to the saline group were treated with saline (~1 µL/15 g body weight) (*n* = 6; 3 males and 3 females), and mice assigned to the G2-β-CD group were treated with G2-β-CD (21.4 µmol/kg) (*n* = 6; 3 males and 3 females). A single cerebroventricular injection of G2-β-CD solution or saline into 4-week-old *Npc1*^−/−^ mice was performed using a stereotaxic instrument for small animals and then motor function was measured every week. We evaluated motor function using the coat hanger and cage lid tests, according to the methods reported by Brooks et al. [21] and Maue et al. [22] with minor modifications. In brief, we used a coat hanger (3 mm diameter, 45 cm length) and the mice were placed at the middle of the horizontal bar with their limbs hanging. The mice were kept at approximately a 30 cm height from a padded surface and maximum hanging time was noted for a maximum of 30 s. For the cage lid test, mice were placed at the center of a mouse cage lid (33 cm length, 22 cm width), which was then inverted. It was placed approximately 30 cm above a padded surface. The time the mouse grasped the inverted cage lid without falling was evaluated as an indication of grip strength. The length of time for the analysis of the cage lid test was 60 s for each mouse.

### 4.8. Statistical Analysis

Statistical analysis was performed using GraphPad Prism (ver. 5.01; GraphPad Software, San Diego, CA, USA). Multiple comparisons were performed to examine the statistical significance of the results. When uniform variance of the results was identified by Bartlett’s analysis (*p* < 0.05), one-way ANOVA was used to test for statistical differences. When significant differences (*p* < 0.05) were identified, the results were further analyzed by Tukey’s (Tukey–Kramer) multiple range test for significant differences among the values. If uniform variance of the results was not identified, non-parametric multiple comparisons were performed. After confirming significant differences (*p* < 0.05) using Kruskal–Wallis analysis, the differences were then examined by applying Dunn’s multiple test. For comparisons of two unpaired values, the unpaired Student’s *t*-test or Mann–Whitney *U*-test was performed.

## Figures and Tables

**Figure 1 ijms-20-01152-f001:**
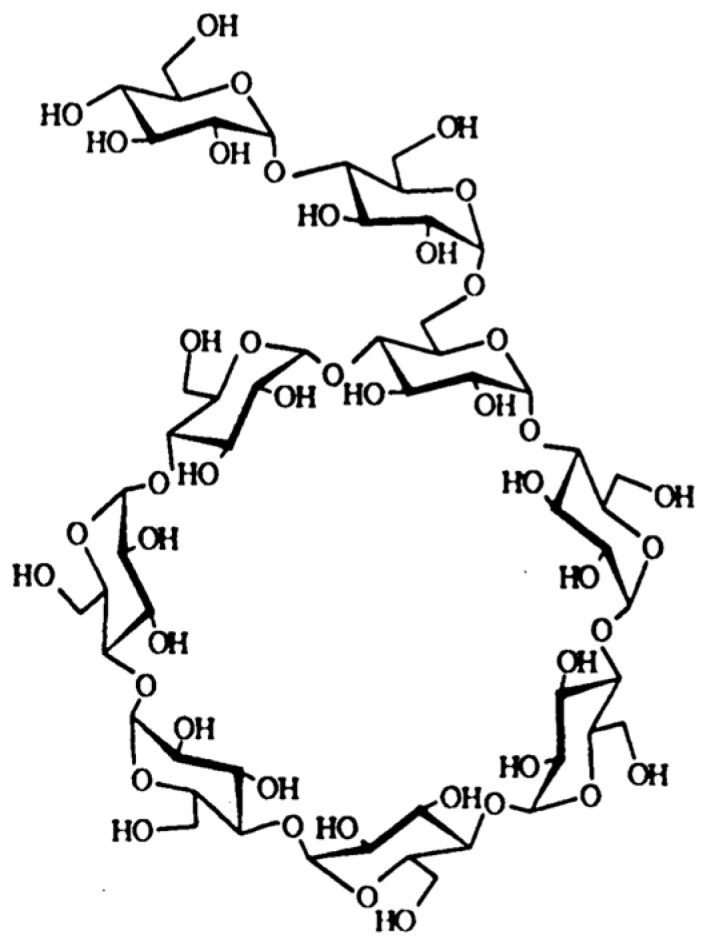
Chemical structure of G2-β-CD.

**Figure 2 ijms-20-01152-f002:**
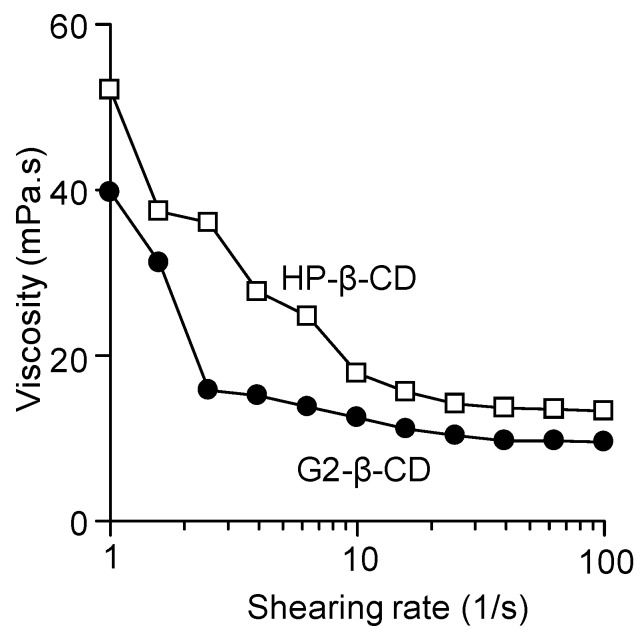
Viscosities of G2-β-CD (300 mM) and HP-β-CD (300 mM) solutions. The viscosities of G2-β-CD (closed circle) and HP-β-CD (open square) solutions were plotted as a function of shear rate.

**Figure 3 ijms-20-01152-f003:**
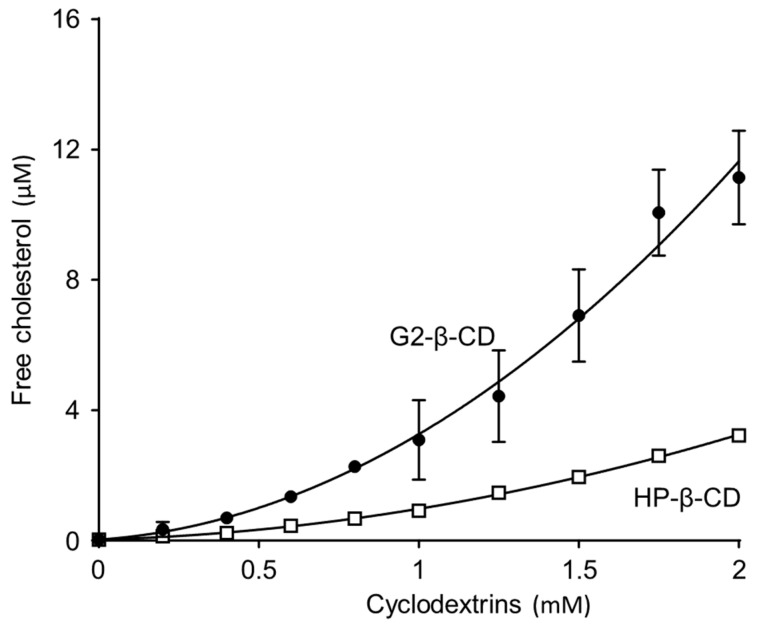
Solubility-curve analysis to evaluate the solubilizing ability of G2-β-CD (closed circle) and HP-β-CD (open square) with free cholesterol (FC) in culture media. The solubility of FC was measured in DMEM/F12 medium (1:1) at 37 °C. Each point represents the mean ± S.D. (*n* = 3–5).

**Figure 4 ijms-20-01152-f004:**
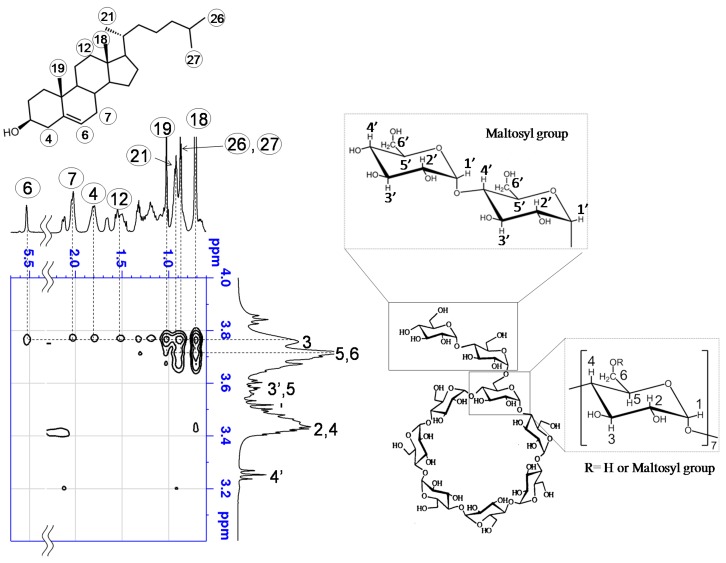
Two-dimensional ^1^H-NMR Rotating frame Overhauser effect spectroscopy (ROESY) spectrum of the G2-β-CD and FC solution. G2-β-CD (100 mM) and FC was dissolved in D_2_O.

**Figure 5 ijms-20-01152-f005:**
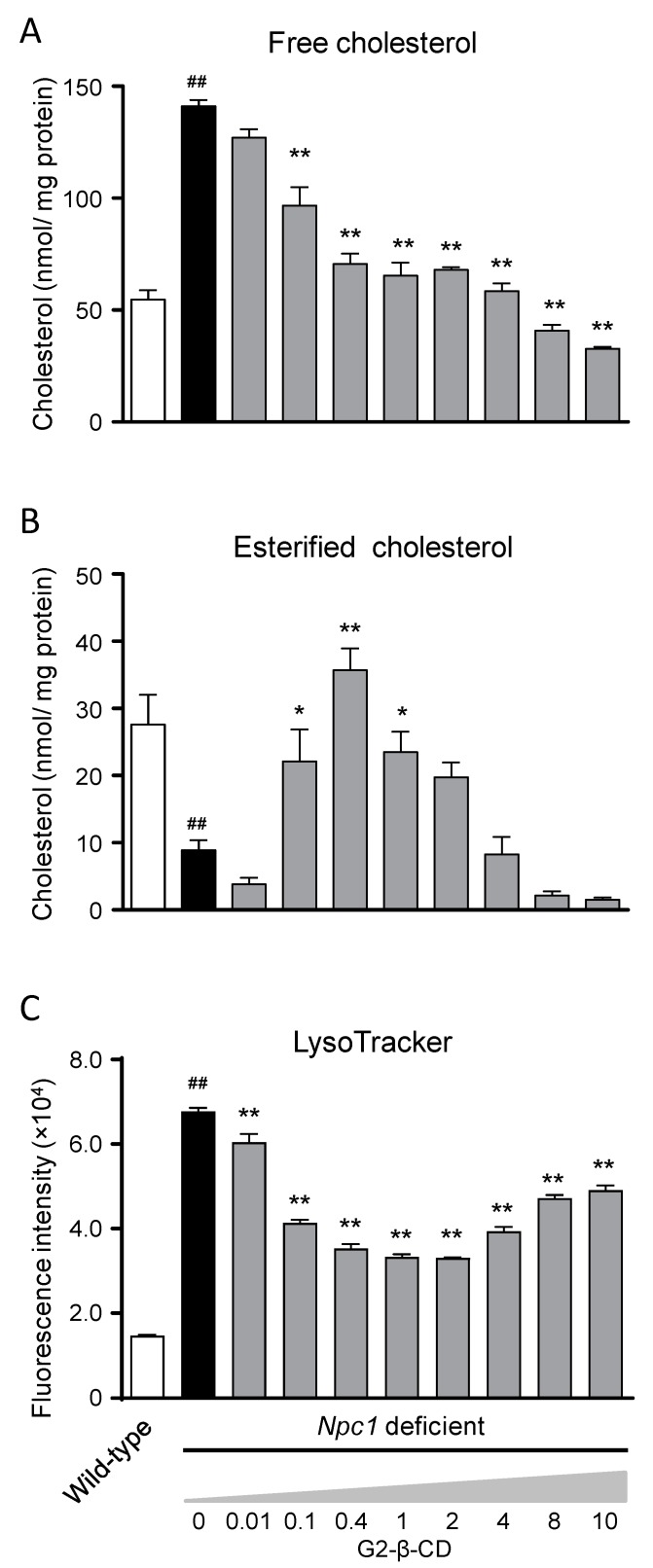
Concentration-dependent effects of G2-β-CD on intracellular cholesterol trafficking and lysosome volume in *Npc1* deficient cells. Intracellular FC (**A**) and EC (**B**) levels were measured 24 h after G2-β-CD treatment of *Npc1* deficient cells. The fluorescence intensity of LysoTracker^®^ was determined by flow cytometry 24 h after G2-β-CD treatment (**C**). Each bar represents the mean ± S.E.M. (*n* = 6 for FC and EC, and *n* = 3 for LysoTracker^®^ experiments). ^##^
*p* < 0.01 compared with the Wild-type group. * *p* < 0.05, ** *p* < 0.01 compared with the 0 mM group.

**Figure 6 ijms-20-01152-f006:**
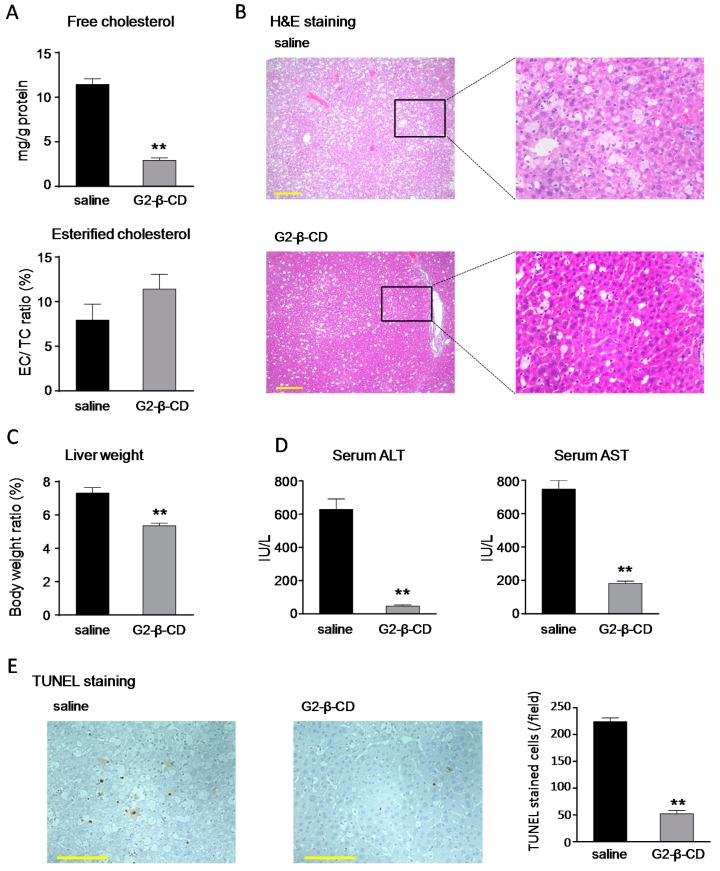
Attenuating effects of systemic G2-β-CD treatment on liver manifestations in *Npc1*^−/−^ mice. Mice were subcutaneously treated with G2-β-CD (2.9 mmol/ kg) from 6 to 8 weeks of age, and then serum and liver samples were collected at 8 weeks and 2 days of age. Free cholesterol (FC) content and percentage of esterified cholesterol (EC) in liver tissue (**A**), liver histological analysis (H&E) (**B**), Liver/body weight ratio (**C**), serum alanine aminotransferase (ALT) and aspartate aminotransferase (AST) levels (**D**) and hepatic DNA fragmentation (TUNEL staining) (**E**) were measured. Scale bar: 200 and 100 μm for H&E and TUNEL staining, respectively. Each bar represents the mean ± S.E.M. (*n* = 8). ** *p* < 0.01 compared with the saline group.

**Figure 7 ijms-20-01152-f007:**
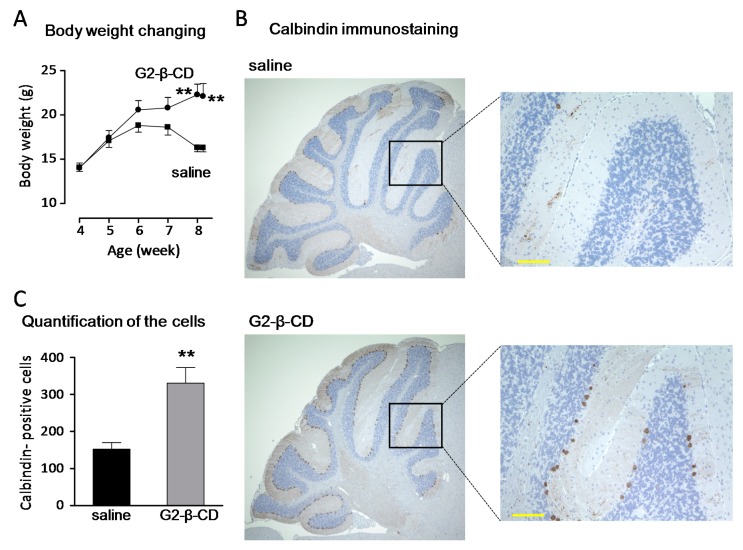
Preventive effects of an intracerebroventricular injection of G2-β-CD on body weight loss (**A**) and Purkinje cell loss in the cerebellum (B and C) of *Npc1*^−/−^ mice. An injection of G2-β-CD 21.4 µmol/kg was performed in mice at 4 weeks of age. The body weight was measured once a week until histological analysis. A brain sample was collected at 8 weeks and 2 days of age and immunohistochemical staining of calbindin was performed. The representative pictures were shown (**B**). Scale bar: 100 μm. The number of calbindin-positive cells was measured (**C**). Each point and bar represents the mean ± S.E.M. (*n* = 5–6). ** *p* < 0.01 compared with the saline group.

**Figure 8 ijms-20-01152-f008:**
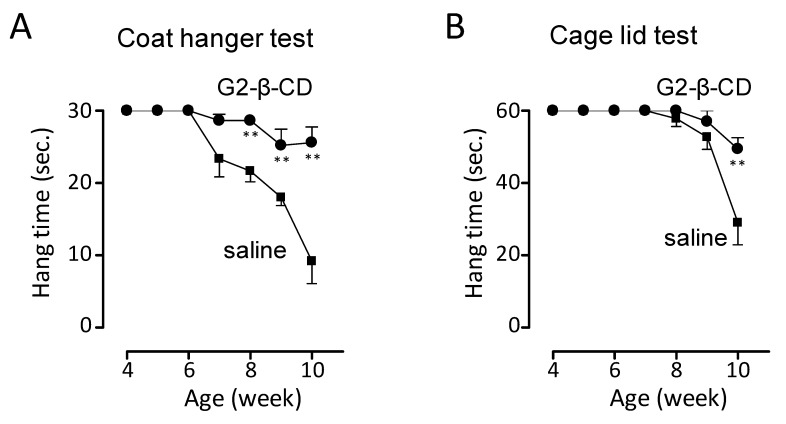
Effect of the intracerebroventricular injection of G2-β-CD on motor function in *Npc1*^−/−^ mice. An injection of G2-β-CD (21.4 μmol/kg) was performed in mice at 4 weeks of age and motor function was evaluated every week up to 10 weeks. Results of the hanging time in the coat hanger test (**A**) and cage lid test (**B**). Each point represents the mean ± S.E.M. (*n* = 6). ** *p* < 0.01 compared with the saline group.

**Table 1 ijms-20-01152-t001:** Surface tensions of G2-β-CD and HP-β-CD solution (300 mM in ultra-pure water) measured by the du Nouy method.

	Ultra-Pure Water	G2-β-CD	HP-β-CD
Surface tension (mN/m)	70.2	68.5	57.6

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
