# Peer review of "In Vitro and In Vivo Evaluation of 6-O-α-Maltosyl-β-Cyclodextrin as a Potential Therapeutic Agent Against Niemann-Pick Disease Type C"

_ijms, 2019, doi:10.3390/ijms20051152_

Round 1
Reviewer 1 Report
The revised manuscript has been supplemented with several new experiments that enriched the contents mostly of the biological evaluation of G2-β-CD.
The previously requested improvement in the processing of the 2D ROESY NMR spectra was not performed, although new figures were included in the supplementary section. However, I consider this task as very simple to implement. Please include a nicer Figure 4 (increase the number of contours!). Apparently the authors were mostly engaged with the supplementation and rationalization of the biological results.
It is a formidable task to prove for G2-β-CD beyond doubt and within a single article biological efficacy, pharmaceutical suitability, mechanism of action even at molecular level. What is presented now, I think is enough.
Therefore I consider the revised manuscript acceptable for publication if the new Figure 4 is supplied and the authors reduce the new discussion paragraph on p.12-13 using less self-criticism and fewer assumptions for the action of G2-β-CD at the molecular level at various concentrations.
Author Response
Reply to the reviewer 1
We thank you for the critical comments and useful suggestions that have helped us to improve our manuscript. We have taken all of these comments and suggestions into account in the revision of our manuscript. In the revised manuscript, the changes are marked in red. Inserted and deleted sentences are pointed out by underline and strike-through, respectively.
Comment
The previously requested improvement in the processing of the 2D ROESY NMR spectra was not performed, although new figures were included in the supplementary section. However, I consider this task as very simple to implement. Please include a nicer Figure 4 (increase the number of contours!). Apparently the authors were mostly engaged with the supplementation and rationalization of the biological results.
Therefore I consider the revised manuscript acceptable for publication if the new Figure 4 is supplied and the authors reduce the new discussion paragraph on p.12-13 using less self-criticism and fewer assumptions for the action of G2-β-CD at the molecular level at various concentrations.
Response to the comment
Thank you very much for your valuable advice. We reanalyzed 2D ROESY NMR spectra and tried to prepare nicer spectra which increased the number of contours. We hope the revised Figure 4 would comply with your request.
In addition, we have condensed and revised the discussion about the limitation of this study and the speculative mechanisms of G2-β-CD. Besides, we have deleted the additional figure 8. The improved discussion was shown in page 13-14.
Reviewer 2 Report
The resubmission by Yasmin et al. is significantly improved. In particular, the addition of the data in the supplement and the neuromuscular function tests have eliminated some of my more significant concerns. That being said, there are a few areas that still need to be addressed:
1) Figure 6B: In the images provided it is difficult to tell if there is microvesicular or macrovesicular steatosis present. The authors need to provide images with higher magnification so the vacuolated hepatocytes and Kuppfer cells are easier to see. It would be best to use a strategy similar to what was used for the cerebellum images (figure 7B).
2) Supplemental figure 6C: This data seems to be repeated with figure 6D in the main manuscript. I do not think it is useful to include this supplemental data.
3) The new discussion paragraph should be improved. In particular, the authors need to not include so much speculative information on potential mechanisms. As mechanisms, such as autophagy, were not investigated in any manner of this report, this information is not needed. This also applies to supplemental figure 8. A working model of the data presented would be much more valuable than a hypothetical mechanism of action for G2-B-CD.
Author Response
Reply to the reviewer 2
We thank you for the critical comments and useful suggestions that have helped us to improve our manuscript. We have taken all of these comments and suggestions into account in the revision of our manuscript. In the revised manuscript, the changes are marked in red. Inserted and deleted sentences are pointed out by underline and strike-through, respectively.
Comment 1
Figure 6B: In the images provided it is difficult to tell if there is microvesicular or macrovesicular steatosis present. The authors need to provide images with higher magnification so the vacuolated hepatocytes and Kuppfer cells are easier to see. It would be best to use a strategy similar to what was used for the cerebellum images (figure 7B).
Response to the comment 1
We are grateful to the reviewer for bringing this to our notice. As reviewer pointed out, we have added higher magnification pictures of liver histology and rearranged the position of each picture and figure in the revised figure 6.
Comment 2
Supplemental figure 6C: This data seems to be repeated with figure 6D in the main manuscript. I do not think it is useful to include this supplemental data.
Response to the comment 2
According to your comment, we have deleted the supplemental figure 6D about transaminase levels.
Comment 3
The new discussion paragraph should be improved. In particular, the authors need to not include so much speculative information on potential mechanisms. As mechanisms, such as autophagy, were not investigated in any manner of this report, this information is not needed. This also applies to supplemental figure 8. A working model of the data presented would be much more valuable than a hypothetical mechanism of action for G2-B-CD.
Response to the comment3
The reviewer made very important point. We have condensed and revised the discussion about the potential mechanisms of G2-β-CD. We have also deleted the speculation about autophagy and additional figure 8. The improved discussion was shown in page 13-14.